# FedORION: Aggregation-Assisted Proxyless Distillation for Heterogeneous Federated Learning

## Abstract

System heterogeneity in Federated Learning (FL) is commonly dealt with knowledge distillation by combining the clients' knowledge via distillation into a global model. However, such knowledge transfer to the global model is often limited by distillation efficiency and unavailability of the client data. Most of the existing approaches require proxy data on the server side for distillation, which becomes a bottleneck. To circumvent these limitations, we propose a novel FL framework, FedORION (Aggregation-Assisted Proxyless Distillation for Heterogeneous Federated Learning) that comprises of deep mutual learning (DML) at client end, and global aggregation followed by noise engineered data-free distillation at the server end. DML enables server side global aggregation which otherwise is infeasible due to different client model architectures. The aggregation results in knowledge integration which is further boosted by the subsequent distillation. This, however, also increases the burden on clients, especially with low computational budget. We, therefore, further introduce the idea of selective mutual learning where only those clients perform DML that are not limited by computational capacity. This reduces the overall computational burden without any compromise in the performance. We conduct rigorous experiments on various publicly available datasets and observe a remarkable improvement in the performance over the existing heterogeneous FL methods. For example, for TinyImagenet dataset, FedORION shows almost three times better performance as compared to the best baseline. The results provide evidence for the utility and effectiveness of our approach and open up a new direction for heterogeneous FL.

## 1 Introduction

Continuous advancement in data driven AI technologies are making the users concerned about the privacy of their personalized data. Federated Learning (FL) has emerged as a solution to the problem which ensures the availability of sufficient data for the development of AI models without raising any privacy concerns. One of the very first FL algorithms was FedAvg McMahan et al. (2016) in which multiple clients participate in collaborative learning of a global centralized model. FedAvg optimizes the global server model by directly averaging the participating clients' model parameters while the data resides with the clients themselves. FedAvg has a limitation of not taking into account clients' heterogeneity Li et al. (2018); Wang et al. (2020), which occurs mainly due to two reasons – statistical differences in data distributed across the clients, known as *data heterogeneity*, and variations in client model architectures and computational capacity, known as *system heterogeneity*. Among the two latter is more challenging as it allows the clients to use a model architecture according to their capacity and makes the task of client model aggregation using a simple technique like averaging infeasible Li & Wang (2019). One possible way to handle system heterogeneity is knowledge distillation (KD) which allows model-agnostic transfer of knowledge from multiple client models to a single global server model Afonin & Karimireddy (2022). However, simultaneous distillation of multiple client models' knowledge into a single server model is naturally difficult. As demonstrated in Afonin & Karimireddy (2022), unlike the FL methods that use direct averaging and its variants, KD-based FL approaches are adversely affected by the mismatch among the client model architectures as well as the mismatch in the data used for training and distillation. There-

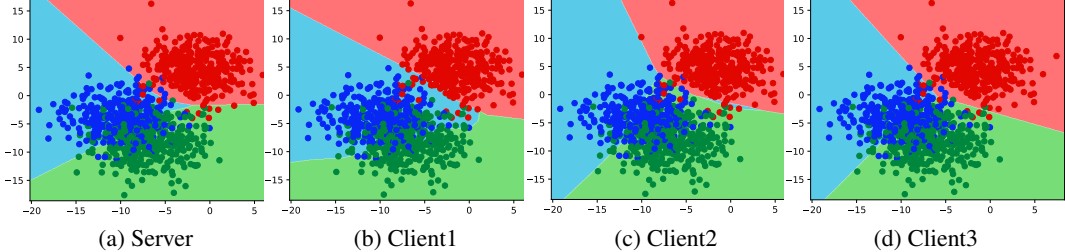

(a) Server      (b) Client1      (c) Client2      (d) Client3

Figure 1: FedORION in action. The figure depicts a toy example of 3 class classification task for samples with just 2 features. We deploy shallow neural networks with only a few fully connected layers and very few neurons per layer. We use the RELU activation function followed by BatchNorm after every layer in the network except the last layer. Each client has a different architecture and learns a different decision boundary, which is then distilled onto the server model. The decision boundaries of the server show the effectiveness of our approach (FedORION).

fore, knowledge integration at the server solely using KD becomes a bottleneck. Moreover, since the server does not have access to the clients' data these approaches either depend on proxy data or synthetic data generated by a pre-trained generator for distillation Lopes et al. (2017). It is a critical dependency as the dissimilarity of proxy/synthetic data with client data hampers the performance.

To alleviate the aforementioned problems, we propose a novel FL framework (abbreviated as FedORION) that addresses the challenge of heterogeneity among the client models using aggregation-assisted distillation while removing the dependency on the proxy/synthetic data. FedORION allows clients and server to have a model of their choice. The server model is shared across the clients and knowledge transfer happens at both ends where local data is used at the client end and noise samples are used at the server. The entire training is divided in three phases - deep mutual learning (DML), knowledge aggregation, and proxyless distillation. DML enables local transfer of knowledge to a common model which is then aggregated at the server and then boosted by the distillation. In particular noise based distillation is inspired from the recent attempts of noise-based learning Raikwar & Mishra (2022); Baradad Jurjo et al. (2021). It uses noise samples such as Gaussian noise for data-free distillation and completely preserves the privacy of the client's data. Consider an illustration of FedORION using a toy problem of classifying 2D points in three classes, shown in Figure 1. The setup contains three clients with each having a different multi-layer perceptron (MLP) model. Each client's MLP learns a different decision boundary using the available local data, also shown in Figure 1. The server uses samples created using Gaussian noise ($\mathcal{N}(0, 1)$) to distill clients' knowledge in the global server model and beautifully adapts its decision boundary for the given task, despite the presence of heterogeneity among the client models. This is an interesting behavior that we describe in detail in section 3 and 4. We perform experiments with different public datasets and observe a robust and better performance as compared to the recently reported relevant approaches. In this work our major contributions are as follows:

- We design a novel FL framework (FedORION) for addressing the client's system heterogeneity that allows clients to have different neural network models at their end and still aggregates the clients' knowledge at the server to deliver a generalized global model.

- The proposed selective DML component in FedORION allows the clients with low computational budget to skip DML without compromising the overall performance, thereby making the method fast and computationally feasible.

- Our approach eliminates the requirement of proxy data and pre-trained synthetic data generating model, which is critical for the KD-based existing heterogeneous FL approaches. Moreover, our experiments on relatively higher level of system heterogeneity than what have been explored so far reveal the limitations of the existing approaches and demonstrate the robustness of the proposed approach.

- Our extensive experiments on publicly available datasets show remarkable performance improvement over the recently reported heterogeneous FL approaches and open up a new direction for heterogeneous FL.

## 2    RELATED WORK

Initial FL approaches mainly focused on aggregation schemes which could only support homogeneous client models. For example, FedAvg McMahan et al. (2016) learns the global server model by directly averaging the clients' model parameters and then broadcasts the global aggregated server model to the clients for further optimization on their private data. Similarly, FedProx Li et al. (2018), which is an extension to the FedAvg, adds a regularizer term for stable and fast training of the client models. SCAFFOLD Karimireddy et al. (2019) is another interesting approach that involves a control variate term for the optimization at the client side and performs global aggregation of the individual client model parameters at the server side. SCAFFOLD is effective in circumventing the data heterogeneity problem but is limited to homogeneous client models. There are several other approaches (FedDyn Acar et al. (2021), FedNova Wang et al. (2020), FedMLB Kim et al. (2022), FedGKD Yao et al. (2021), FedNTD Lee et al. (2022) etc.) that are based on the same idea of aggregating the clients' model parameters on the server side. The major disadvantage of these approaches is their inability to address clients' model heterogeneity as the aggregation requires model architectures to be identical.

Some recent methods that handle clients' model heterogeneity utilize the idea of KD for transferring the knowledge from client models to the server model or helping client models to learn representations of their peer client models. KD Hinton et al. (2015) refers to the transfer of knowledge from one neural network (teacher model) to another (student model) using soft labels. Conventional KD requires the same dataset on which the teacher model was trained to be available for distillation. Usually, in FL, access to clients' data is prohibited, thus data-free KD Lopes et al. (2017) provides a way forward and preferred by the approaches targeting clients' model heterogeneity. For example, FedDF Lin et al. (2020) proposed robust model fusion which aggregates similar client models at the server end and creates multiple global server models. It then uses KD to transfer the clients' knowledge into the aggregated global models using average logits. FedDF requires a global proxy dataset for distillation as the original data resides only with clients. Similarly, FedMD Li & Wang (2019) also uses KD, however, it does not learn a global server model but distills the knowledge into clients themselves using a global proxy dataset and average logits. The overall performance is evaluated by averaging the individual client's accuracy. In contrast, FedKEMF Yu et al. (2022) performs mutual knowledge transfer at the client side and ensemble knowledge transfer at the server side. FedKEMF also requires a global proxy dataset for distillation. Similarly, DS-FL Itahara et al. (2023) utilizes the concept of KD and learns the global server model by average logits using a proxy dataset. FD-meta Liu et al. (2022) uses KD to learn a global server model. There are various other approaches that address model heterogeneity by utilizing proxy datasets, for example, CRONUS Chang et al. (2019), Kt-pFL Zhang et al. (2021) etc. All these approaches are prone to failure in absence of the relevant proxy data. A possible alternative to the proxy dataset can be a pre-trained generator which is responsible for the generation of synthetic data for KD. However, to the best of our knowledge, the pre-trained generator is exploited only for homogeneous clients' models. The approaches which utilize pre-trained generator includes FedFTG Zhang et al. (2022), FedGEN Zhu et al. (2021). We propose to eliminate this dependency on the publicly available proxy data by using Gaussian noise samples for performing data-free KD without compromising on the performance.

In contrast to KD-based approaches, there is another set of heterogeneous FL methods which include FJORD Horvath et al. (2021), HeteroFL Diao et al. (2020), FedRolex Alam et al. (2022), and FLANC Mei et al. (2022) etc. These are primarily model pruning methods which means the clients do not have the flexibility of choosing a model. Server has a large global model and the clients' models are the pruned subnets of the global model. These methods technically support system heterogeneity as they allow clients to have different model architectures, but they do not allow the clients to choose the model of their choice. In contrast our approach is comparatively less restrictive in terms of model choice.

## 3    METHOD: FEDORION

In this section, we provide a detailed description of our approach (FedORION). We mainly distribute our approach into three phases. 1) DML phase at the client side, 2) Global aggregation phase at the server side, 3) Proxyless distillation phase at the server side. We further discuss each of these in the following subsections.

### 3.1 DEEP MUTUAL LEARNING (DML) PHASE

KD-based existing heterogeneous FL approaches needs to simultaneously address the problems arising due to mismatch among the clients' models and mismatch between the server's proxy data and clients' training data Afonin & Karimireddy (2022). Through the DML phase, FedORION aims to decouple these problems. DML Zhang et al. (2017) is based on KD Hinton et al. (2015) in which different deep learning models participate together and for each model (student model), the rest of the other models act as teacher model, and similarly, all the model get trained. For our approach, we consider only two architectures for DML at the individual client. These two architectures include the global server model and the local client model. In this case, each model learns from the local data and additionally acts as a teacher for the other model. The data mismatch problem is implicitly solved as only the client data is used. Both, global and local models try to optimize their parameters using the loss functions described below.

Let us assume $\theta_g$ as the global server model parameters and $\theta_i$ as the $i^{th}$ client model parameters. The server broadcasts copies of the global server model to individual clients. $\theta_{gi}$ represents the copy of global model at the $i^{th}$ client. The client $i$ holds the dataset $D_i$ which is a set of individual data points $(x_j, y_j)$ where $x_j$ and $y_j$ indicate the $j^{th}$ data point and target class respectively. For $i^{th}$ client, let us assume that neural network models are realized as $f(\theta)$. Accordingly $f(x; \theta_{gi})$ and $f(x; \theta_i)$ represent the output of the global server model copy and the local client model for input sample $x$, respectively. The mutual learning loss for the client and server model is as follows.

$$L_{CMi} = L_{\mathcal{XE}}\left(f(x_j, \theta_i), y_j\right) + L_{\mathcal{XE}}\left(f(x_j; \theta_{gi}), f(x_j, \theta_i)\right) \tag{1}$$

$$L_{GMi} = L_{\mathcal{XE}}(f(x_j, \theta_{gi}), y_j) + L_{\mathcal{XE}}(f(x_j, \theta_i), f(x_j, \theta_{gi})) \tag{2}$$

where $L_{\mathcal{XE}}(., .)$ is the standard cross-entropy loss function. The mutual loss for $i^{th}$ client and the corresponding global model copy are $L_{CMi}$ and $L_{GMi}$, respectively. After training global and local models mutually on the client's private data, these two models are then communicated to the server for aggregation and distillation at the server side. A side benefit which DML provides is that it allows the global model to get exposed to the client's data.

### 3.2 GLOBAL AGGREGATION PHASE

Once the server receives the local models and copies of the global model from different clients, it aggregates the global model copies ($\theta_{gi}$) by directly averaging their parameters. DML enables local knowledge transfer in the global model copies which are identical in terms of architecture and can, therefore, be directly aggregated resulting in an aggregated global model ($\theta_g$) which is then utilized in the next phase for distillation. The advantages of direct averaging during aggregation phase of FedORION come at the cost of additional computational burden for the training of a global model copy during the DML phase. Particularly for the clients with low computational capacity, performing DML using two models may not be feasible. We, therefore, introduce the idea of selectively performing DML in FedORION, which is described in the following subsection.

#### 3.2.1 FEDORION WITH SELECTIVE DML

FedORION with Selective DML allows the clients with limited computational capability to skip DML. Instead, these clients are all assigned the smallest model (in terms of number of parameters) among the available choices of models and the same model is also used as the global model. This ensures that even when some of the clients do not perform DML, the global aggregation phase remains intact. Client models from those which do not perform DML are combined with global model copies from those which do perform DML using direct averaging to produce the aggregated global model. We later show in experiments that even though a lightweight global model is used, FedORION with selective DML is still able to achieve remarkable performance as compared to baseline methods.

### 3.3 PROXYLESS DISTILLATION

Although the global aggregation phase allows the knowledge of global model copies to be combined into a single global model, we attempt to boost its representation by using the locally trained client models. This is done while considering the fact that in the heterogeneous environment the

---

**Algorithm 1** FedORION

---

**In Client:**
**Input:** Client's local model $\theta_i$, Copy of global server model $\theta_{gi}$, Number of epochs $T$, Private dataset $D_j$, Learning rate for the global model $\eta_g$, Learning rate for local model $\eta_i$
**for** each epoch $r = 1, 2, ..., T$ **do**
    **for** each batch $B = \{(x_j, y_j)\}_j \in D_j$ **do**
        $\theta_i = \theta_i - \eta_i \nabla_{\theta_i} L_{CMi} \triangleright$ eq. (1)
        $\theta_{gi} = \theta_{gi} - \eta_g \nabla_{\theta_{gi}} L_{GMi} \triangleright$ eq. (2)
    **end for**
**end for**
**return** $\theta_i, \theta_{gi}$

**In Server:**
**Input:** Client's local models $\{\theta_i\}_{i=1}^C$, Copy of global server models $\{\theta_{gi}\}_{i=1}^C$, Number of epochs $T$, Learning rate for the global model $\eta_g$
Aggregation $\theta_g = \frac{1}{C} \sum_{i=1}^C \theta_{gi}$
**for** each epoch $r_s = 1, 2, ..., T_s$ **do**
    Sample a batch of noise $z(\sim \mathcal{N}(0, 1))$
    Use current batch-norm statistics
    $\theta_g = \theta_g - \eta_g \nabla_{\theta_g} L_{KD} \triangleright$ eq. (3)
**end for**
Broadcast $\theta_g$ to the clients

---

learning dynamics of individual client models are affected by the scale of the local data and inductive bias due to the choice of model architecture. We use data-free KD to transfer the knowledge from different client models to the aggregation global model. This phase is similar to the existing KD-based heterogeneous FL approach, however a big difference is that we use noise samples instead of proxy data for the distillation. We take inspiration from Raikwar & Mishra (2022) which reported an effective method to perform KD using Gaussian noise samples. The key idea is to utilize current statistics of the BatchNorm layers instead of using running statistics. A bottleneck in such noise-based distillation is the shift in the distribution of hidden layer activation maps of the teacher model which is caused by feeding Gaussian noise instead of the expected original data. When we use current statistics in BatchNorm layers of the teacher model, it normalizes the activation maps according to the input data distribution to mitigate the shift which is not the case with the statistics (running) derived from the original training data. Accordingly, in FedORION, client models, which are trained on the data residing only at the clients, act as teacher for the aggregated server model and the server distills the client knowledge using Gaussian noise instead of proxy data. Note that the normalization of activation maps also reduces the adverse impact of mismatch between clients' data and the samples used by the server. An illustration of the entire workflow of FedORION is provided in appendix A. The loss function used by server for KD ($L_{KD}$) is as follows.

$$L_{KD} = \sum_{i=1}^C \frac{n_i}{\sum_{i=1}^C n_i} L_{\mathcal{XE}}(f(z; \theta_i), f(z; \theta_g)) \tag{3}$$

$f(z, \theta_i)$ and $f(z, \theta_g)$ represent the outputs of $i^{\text{th}}$ client model and aggregated server model on Gaussian noise sample ($z \sim \mathcal{N}(0, I)$), respectively. $n_i$ denotes the total number of samples of private data at the $i^{th}$ client. Functioning of FedORION is shown in Algorithm 1.[1] Note that, the noise-based distillation not only allows knowledge transfer from heterogeneous clients but also provides flexibility to the server to create as many noise samples for KD as required.

## 4 EXPERIMENTS AND RESULTS

In this section, we discuss experiments and present the obtained results. We consider six recent heterogeneous FL approaches as baselines for comparison with FedORION. These include KT-pFL Zhang et al. (2021), FedMD Li & Wang (2019), FedDF Lin et al. (2020), DS-FL Itahara et al. (2023), FedKEMF Yu et al. (2022), and FDmeta Liu et al. (2022). All these use KD for efficient knowledge transfer and are therefore the most relevant baselines for our method. As the codes for the considered baselines are unavailable, we implement them ourselves by closely following the details provided in the corresponding articles. [2].

---

[1]The codes are provided in supplementary material.

[2]For completeness, the results for comparison with FedAvg McMahan et al. (2016), FedProx Li et al. (2018), SCAFFOLD Karimireddy et al. (2019), and FedNTD Lee et al. (2022) are also provided in appendix D

## 4.1 Dataset and Experimental Settings

We consider five different lightweight deep learning architectures namely Resnet18, Resnet34, ShuflenetV2, MobilenetV2, and WRN40 (Wide Resnet 40) to get sufficient client model heterogeneity. Unless specified, we use WRN40 as the global server model for all the methods except for FedMD, and KT-pFL, which do not learn the global server model at all. In our experiments, each client is randomly assigned one of the considered models. The criteria for selecting clients which do not perform DML is simple. We randomly initialize the clients' models and the clients which have WRN40 as their local model do not perform DML. For all methods, we use Stochastic Gradient Descent (SGD) technique with a momentum of 0.9 for both the client and server side optimization. The learning rates for clients' and server are set to 0.01 and 0.001 respectively in all the experiments. The batch size on the client side is always kept to 32. For the baselines and our method (FedORION) we use a batch size of 128 as a relatively higher batch size is needed to get reasonable performance in noise-based KD Raikwar & Mishra (2022). We take number of clients ($N$) $\in$ {30, 20, 100} and Dirichlet parameter ($\beta$) $\in$ {0.6, 0.1, 0.01}. These values are commonly found across FL literature eg. FedDYN sets $\beta$ to 0.6, FedDF/KT-pFL uses 21/20 clients, FedKEMF uses 30 clients. The settings including $N = 100$, $\beta = 0.1$ are more practical scenarios and are followed in lots of FL papers. We evaluate the performance of all the methods by using test set accuracy of the global server model on the test set except for FedMD, and KT-pFL which reports the average of individual accuracy of all the client models on the test set. FedDF also does not learn a single global model but learns a set of global models. Its performance is, therefore, reported based on the ensemble of the global models. We calculate the ensemble by averaging the output logits followed by softmax layer to get final prediction probabilities. We extend the observations of FedORION on toy example (Figure 1) for comparatively more realistic tasks involving higher number of classes and data points, and present results in following subsections. We perform experiments on CIFAR10, CIFAR100, and TinyImagenet datasets for non-IID settings including model and data settings. We describe the experiments on each dataset below.

## 4.2 Experiments on CIFAR10

For CIFAR10, we use CIFAR100 as the proxy dataset for all the baselines. We perform experiments both on varying model and data heterogeneity, we run all the approaches for 100 rounds, with each round involving 10 local epochs for clients and 5 epochs for the server. The specific settings for varying model and data heterogeneity are mentioned as follows.

### 4.2.1 Experiments with varying model settings

For all experiments, we use total 30 clients where 30% of the randomly sampled clients participate in each round. We only vary model heterogeneity for this experiment. We kept fixed the heterogeneity in data by using Dirichlet distribution with $\beta$ denoting the Dirichlet parameter which is set to 0.6, where lower $\beta$ results in higher data heterogeneity. We perform three types of experiments which include homogeneous (Hom), lower heterogeneity (L-Het), and higher heterogeneity (H-Het) models. The methods supporting model heterogeneity can be trivially extended to homogeneous model setup. The homogeneous model setup uses only WRN40 model as the global model which is identically distributed across all the clients. For L-Het we use only three models i.e Resnet20, Resnet32, ShufflenetV2 which are same as the FedDF's model heterogeneity. For FedORION, DML in case of L-Het is not performed at the clients having Resnet20. The H-Het setting includes all five considered models, as described earlier in section 4.1. Table 1 shows results for CIFAR10 with non-IID data settings and varying levels of model heterogeneity. FedORION shows the best performance in terms of test accuracy as compared to other baselines. FedDF, on homogeneous settings, shows a comparable performance with FedORION. However, FedORION outperforms its most competitive baseline, FedDF in L-Het, and H-Het model settings by a considerable margin. Note that FedORION maintains its performance across different levels of model heterogeneity. However, on L-Het, the results are somewhat less intuitive as the accuracy is lower than H-Het setting. We believe this behaviour is due to the comparatively lower capacity models chosen for L-Het. Furthermore, Figure 2(a,b,c) shows the learning behavior of all methods for Hom, L-Het, and H-Het model settings. The superior and stable convergence of FedORION is evident from these plots.

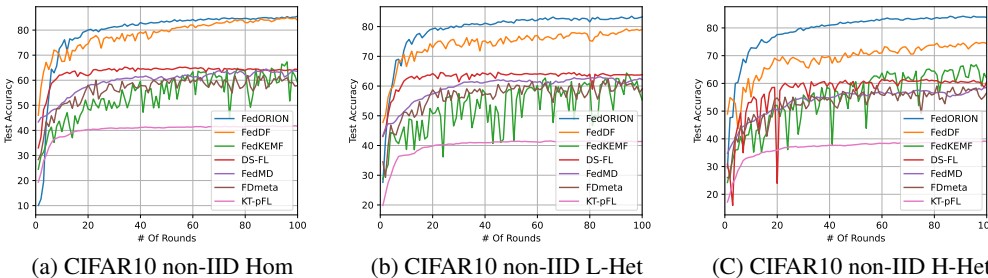

| (a) CIFAR10 non-IID Hom | (b) CIFAR10 non-IID L-Het | (C) CIFAR10 non-IID H-Het |

Figure 2: Learning behavior of FedORION and baselines in the Hom, L-Het, H-Het settings for CIFAR10. As the complexity of the task starts to increase from left to right, the baseline methods suffer a degradation in their learning process. However, FedORION shows a consistent learning behaviour, depicting the generalizability and stability in varying levels of model heterogeneity.

Table 1: Comparative performance of FedORION and baselines on CIFAR10 under non-IID data setting and varying levels of model heterogeneity. The numbers reported are on test set of CIFAR10. Hom denotes homogeneous model setting, and L-Het and H-Het denote lower and higher model heterogeneity, respectively. Results of three runs are shown in form of mean ± standard deviation.

| Setting | FedMD | KT-pFL | FedDF | DS-FL | FDmeta | FedKEMF | FedORION |
|---------|-------|--------|-------|-------|--------|---------|----------|
| Hom | 64.5 ±0.16 | 43.0 ±0.85 | **84.8 ±0.24** | 64.9 ±0.84 | 63.0 ±1.16 | 67.9 ±0.63 | **84.5 ±0.71** |
| L-Het | 62.8 ±0.29 | 42.6 ±0.86 | 77.8 ±1.14 | 64.8 ±0.21 | 63.5 ±0.79 | 65.0 ±1.74 | **82.5 ±0.82** |
| H-Het | 58.4 ±2.21 | 38.6 ±0.43 | 74.5 ±0.17 | 62.1 ±0.52 | 55.5 ±5.61 | 68.0 ±0.85 | **84.2 ±0.33** |

### 4.2.2 EXPERIMENTS WITH VARYING CLIENTS AND DATA HETEROGENEITY SETTINGS

In these experiments we consider H-Het set of models and vary number of clients ($N$) and $\beta$. In each experiment 10 randomly sampled clients participate in each round. The rest other experimental settings including hyperparameter values are kept same as mentioned in section 4.1. As shown un Table 2, FedORION considerably outperforms all baselines on various level of data heterogeneity. An interesting observation here is, FedORION's performance remains consistent even with increasing data heterogeneity (reducing $\beta$) for 30 clients. For 100 clients, the performance is lower than that of 30 clients, which is mainly due to very few samples per class available at each client. Nevertheless, it still significantly outperforms all baselines including the most competitive baseline, *i.e.* FedDF. We also compare FedORION and FedDF's computational complexity in appendix B and find FedORION to be comparatively more efficient.

### 4.3 EXPERIMENTS ON CIFAR100 AND TINYIMAGENET

Next, we consider a comparatively more complex task than the previous one using CIFAR100 and TinyImagenet datasets. These are 100 and 200-class classification tasks respectively with per class samples considerably smaller than CIFAR10. For CIFAR100 we use CIFAR10 and for TinyImagenet we use CIFAR100 as the proxy dataset for all the baselines. We use 20 clients in total where 50% of the total clients selected at random participate in each round. We introduce heterogeneity in the data using the Dirichlet distribution with $\beta$ set to 0.6, and 0.1. The ratio of selective DML in these experiments is kept same as mentioned in section 4.1 System heterogeneity is realized using all five considered models, as described earlier in section 4.1. We run all the approaches including baselines for 100 rounds for CIFAR100 and 150 rounds for TinyImagenet, with each round involving 10 local epochs of clients and 5 global epochs of the server.

The results for CIFAR100 and TinyImagenet are presented in Table 2. FedORION outperforms the other methods in these complex tasks with comparatively larger margins, reflecting its robustness. Especially, FedORION shows almost three times greater performance as compared to the best baseline, *i.e.* FedMD, in TinyImagenet experiment. Similarly, it is two times better than the best baseline in case of CIFAR100. Not only that, FedORION also shows a stable and generalized performance on varying data settings on CIFAR100 dataset. The TinyImagenet experiment reveals the limitations

Table 2: Comparative performance of FedORION and baselines on CIFAR10, CIFAR100, Tiny-Imagenet (TImg) under non-IID data setting but with varying levels of number of clients ($N$) and Dirichlet parameter ($\beta$) . The numbers reported are on the test set of the respective dataset.

| Dataset | $N$ | $\beta$ | FedMD | KT-pFL | FedDF | DS-FL | FDmeta | FedKEMF | FedORION |
|---|---|---|---|---|---|---|---|---|---|
| CIFAR10 | 30 | 0.6 | 58.5 | 39.2 | 74.7 | 61.4 | 59.3 | 66.8 | **84.2** |
| CIFAR10 | 30 | 0.1 | 53.0 | 28.2 | 69.2 | 59.0 | 59.1 | 14.2 | **84.7** |
| CIFAR10 | 30 | 0.01 | 35.9 | 18.0 | 53.4 | 29.4 | 33.1 | 15.8 | **86.3** |
| CIFAR10 | 100 | 0.1 | 28.7 | 20.6 | 62.8 | 45.3 | 46.4 | 30.3 | **75.4** |
| CIFAR100 | 20 | 0.6 | 19.6 | 13.8 | 29.9 | 22.5 | 11.7 | 24.0 | **59.3** |
| CIFAR100 | 20 | 0.1 | 14.1 | 10.3 | 27.4 | 13.8 | 05.3 | 01.3 | **60.5** |
| TImg | 20 | 0.6 | 13.6 | 02.9 | 01.0 | 04.9 | 01.9 | 01.0 | **43.2** |

Table 4: (Left) Studies using FedORION on CIFAR10 with variations in types of noise samples including Gaussian noise (GN), DeadLeaves (DL) and StyleGAN (SG). (Right)) FedORION performance With varying global model architectures on CIFAR10/100, and TinyImagenet datasets.

| Method | Accuracy |
|---|---|
| GN | 84.2 |
| DL | 82.6 |
| SG | 84.0 |

| Dataset | WRN40 | WRN16 | Resnet20 | Resnet32 |
|---|---|---|---|---|
| CIFAR10 | 81.6 | 79.1 | 80.0 | 81.7 |
| CIFAR100 | 56.9 | 54.1 | 56.0 | 56.1 |
| TinyImagenet | 40.8 | 38.1 | 40.6 | 40.9 |

of existing approaches. Apart from the increased task complexity, choice of proxy data also affects their performances. Unlike the experiments with CIFAR10 and CIFAR100, the similarity between client and proxy data in this case is low, thus, the performance of some of the baselines is as good as the random chance. In contrast, FedORION is invariant of the choice global proxy data as it uses Gaussian noise samples for distillation. Not only that, FedORION is also unaffected by choice of noise samples which is elaborated in the next section.

## 5  ABLATION STUDIES

We perform various ablation studies while considering variation in different components of FedO-RION. We use CIFAR10 dataset for performing the first four studies and consider other datasets for the final two, under the H-Het settings described in section 4.2. The test accuracy corresponding to each study is reported in Table 3 and 4. We discuss these in the following part of the section.[3]

*Effect of DML:* DML helps both the global model and the local model of the clients to learn from each other's knowledge. We alter this step where instead of using the whole global model loss on the client side ($L_{GMi}$ in equation equation 2) we just rely on distillation part of it. This results in less exposure to the true labels for the global model. Thus, the steps afterward suffer the impact and do not collect the desired amount of knowledge hence result in a reduced performance.

*Effect of Global Aggregation:* The global aggregation phase performs aggregation of the global model copies received from the clients by directly averaging their model parameters. We observe a significant degradation in the performance when the aggregation phase is removed completely and the server relies only on distillation. The indirect homogeneity induced by the global server model copies received from clients allows aggregation and helps in reducing the impact of mismatch among the client models. This is the most critical phase which enables integration of knowledge distributed across clients and decreases the dependency on KD.

*Effect of Noise-Based Distillation:* As data privacy is one of the core features of FL, noise-based distillation is a key step that ensures the privacy of the clients' data and also eliminates the need for any other proxy dataset or synthetic data generator on the server side. We observe a considerable amount of reduction in performance when distillation is removed from the server side and only the aggregation of global model copies is performed. The knowledge from participating clients does not transfer sufficiently and the aggregated model results in a suboptimal performance.

*Effect of Different Noise Samples:* Next we study the effect of different noise samples. We keep all the setting and steps similar to original FedORION. However, the primary difference lies in noise-

---

[3]The learning behaviour of FedORION in various ablation studies is provided in appendix C.

Table 3: Ablation studies showing the significance of DML, aggregation (Agg), and proxyless distillation (Dist). Exclusion of any of these results in accuracy (Acc) degradation for FedORION.

| DML | Agg | Dist | Acc |
|-----|-----|------|------|
| × | ✓ | ✓ | 52.5 |
| ✓ | × | ✓ | 28.5 |
| ✓ | ✓ | × | 69.7 |
| ✓ | ✓ | ✓ | 81.6 |

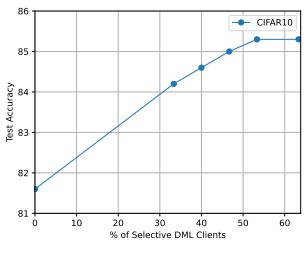
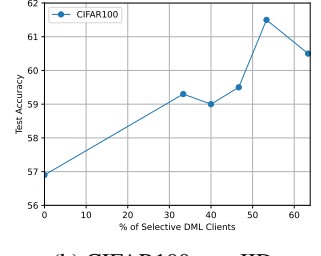

(a) CIFAR10 non-IID    (b) CIFAR100 non-IID

Figure 3: Effect of selective DML on the global model test accuracy. The best performance is observed when $\sim 53\%$ of clients do not perform DML.

engineered distillation phase which now uses samples from DeadLeaves and StyleGAN Baradad Jurjo et al. (2021) instead of Gaussian noise. We observe a limited variation in the performance showing the robustness of FedORION against the variation in data used for server side distillation.

*Effect of selective DML:* To understand the impact of selective DML better, we consider varying its ratio and plot the observations in Figure 3(a) and (b) for CIFAR10 and CIFAR100, respectively. We observe that the test accuracy improves as we decrease the number of clients performing DML. This may sound counter-intuitive, however, we believe the reason behind this behaviour is that as the number of clients not doing DML increases, the homogeneity among the client models also increases and the aggregation step drives the performance. Furthermore, the accuracy either saturates or declines after reaching to a maximum point. This happens because when the ratio of the number of clients not doing DML becomes large, most of the clients are left with the small WRN40 model and the contribution of comparatively larger models (which are likely to learn better) in the knowledge sharing process gets reduced. Therefore, the performance suffers a trade-off in terms of aggregation and knowledge distillation.

*Choice of Global Model:* In this study, instead of WRN40, we use other lightweight models like WRN16, Resnet20, and Resnet32 as the global model. We keep the experimental settings similar to the settings used for the respective datsets in section 4.1. Table 4 highlights an interesting behaviour of FedORION which shows that its effectiveness in distillation of knowledge into the global model is almost insensitive to the choice of model architecture.

## 6 LIMITATIONS AND FUTURE WORKS

FedORION inherits the limitations of noise-based distillation which are discussed in Raikwar & Mishra (2022). Besides as shown in in appendix D it does not perform as good as the homogeneous FL approaches, like FedNTD, under homogeneous settings. This is possibly due to the direct averaging step involved at the server, which can be easily replaced with its sophisticated alternatives available in homogeneous FL literature. This is an interesting future direction to explore and improve the proposed approach. In fact we believe that FedORION, establishes a new paradigm for heterogeneous FL with several possibilities for future.

## CONCLUSION

In this work, we try to address a highly practical scenario in FL, which provides flexibility to the clients to choose a model according to their computational capacity and requirements. We presented a novel FL approach, FedORION, which involves mutual learning at the client side followed by global aggregation and proxyless distillation at the server side. It showed a considerably better performance (in terms of test accuracy) as compared to the other existing KD-based heterogeneous FL approaches. FedORION is found to be robust against the variations in level of heterogeneity (model and data distribution) and dataset used at the server for distillation. Our work reduces the critical dependency of knowledge integration on KD while providing a new way of handling heterogeneity in FL and hopefully will motivate future research in this direction.

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

# Appendix

## A    FEDORION WORKFLOW

The workflow for FedORION indicating Deep Mutual Learning (DML), Aggregation, and Noise base distillation phases is depicted in Figure 4.

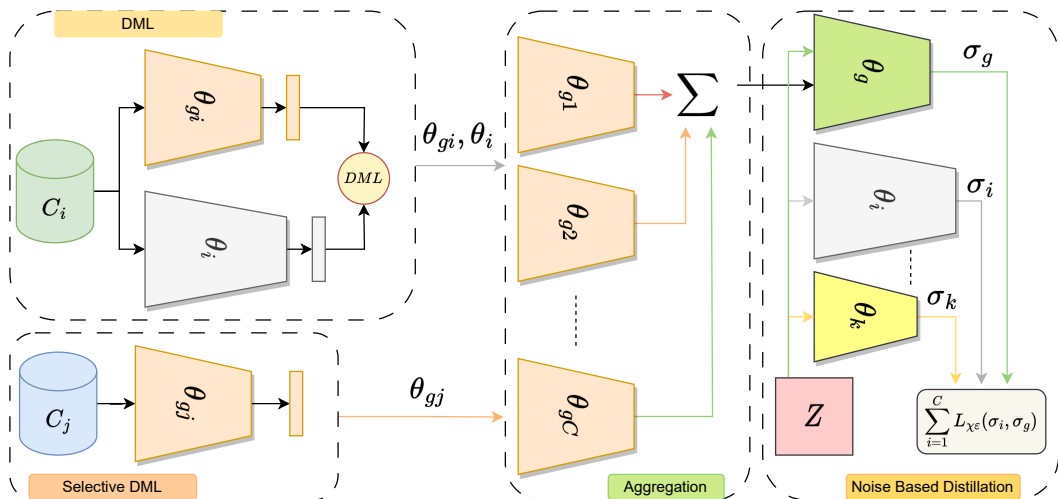

Figure 4: Diagram depicting FedORION's workflow. The clients use mutual learning to simultaneously train the copy of the global server model and its own local model. $\theta_i$ indicates the local model parameters at the $i^{th}$ client and $\theta_{gi}$ indicates the parameters of a copy of global server model received from the server. As depicted, the $i^{th}$ client is performing DML and $j^{th}$ client, due to its computational restrictions selectively choose not to perform DML. the DML phase includes loss, $L_{CMi}$ and $L_{GMi}$ which are presented in equation equation 1 and equation 2 respectively. The global models copies received from clients are first aggregated by direct averaging of the model parameters. The aggregated global model ($\theta_g$) is then used for KD from clients' models using noise samples. $z$ indicates the noise samples that include samples from Gaussian noise. $\theta_{gi}$ indicates parameters of the global model copy received from $i^{th}$ client, $C$ denotes the total number of participating clients. Here $\sigma_i, \sigma_g$ is used as a shorthand for $f(z; \theta_i), f(z; \theta_g)$ respectively. The server loss is $L_{KD}$ and is presented in equation equation 3.

## B    SPEEDUP COMPARISON FEDORION AND FEDDF

We mathematically compare the computation complexity of FedORION with our most competitive baseline, FedDF by showing the speedup gained by FedORION over FedDF.

We take $P$ model prototypes where the largest and smallest models are ResNet18 (11M parameters) and WRN40 (0.5M parameters) respectively. Considering the difference in size, WRN40 takes considerably less time in forward (FP) and backward propagation (BP) as compared to ResNet18.

We consider the following notations

$I_c$: Local clients iterations

$I_s$: Global server iterations

$N$: Number of clients

$P$: Number of unique model prototypes

$M(>1)$: Ratio of computation time of ResNet18 and WRN40

$f_{w40}$: Time taken by WRN40 model for one FP

$f_{r18} :\sim Mf_{w40}$: Time taken by ResNet18 model for one FP

$K(>1)$: Ratio of time taken for BP step as compared to FP

$R_{18}$: Total computation cost of ResNet18 for one FP and BP

$R_{40}$: Total computation cost of WRN40 for one FP and BP

$\alpha$: Ratio of clients having WRN40 model

For our comparison, $I_c = 10$, $I_s = 5$, $P = 5$, $K \sim 3$ We assume all clients participate in every round and the model are uniformly distributed among the clients. We keep WRN40 as the global model.

For WRN40, the computational cost of one FP and BP per iteration is $R_{40} = f_{w40} + Kf_{w40}$, since BP takes $K$ times more time than FP. Similarly for ResNet18, the computation becomes $R_{18} = (K + 1)f_{r18} = M(K + 1)f_{w40} = MR_{40}$.

## B.1 FEDORION

FedORION Performs selective DML at clients, and aggregation followed by noise based distillation at the server. We perform selective DML $100\alpha\%$ of clients since $100\alpha\%$ clients receive the smallest model i.e.WRN40.

### B.1.1 CLIENT

$100\alpha\%$ clients do not perform DML, therefore, their computation would be $\alpha NR_{40}$. For rest of the clients, the computation would be $O((1 - \alpha)N(R_{18} + R_{40}))$. Therefore, the total computation of all clients for $I_c$ iterations becomes,

$$O((\alpha NR_{40} + (1 - \alpha)N(R_{18} + R_{40}))I_c) = O((1 + (1 - \alpha)M)R_{40}NI_c)$$

### B.1.2 SERVER

The server first performs aggregation which involves no FP or BP steps. Therefore, its computation is negligible as compared to the distillation step.

We perform distillation on WRN40 using rest of the clients models. Therefore, for global model, FP and BP both are performed and rest other models perform only FP. The overall computation at the server for $I_s$ iterations would be

$$O((R_{40} + N(\alpha f_{w40} + (1 - \alpha)f_{r18}))I_s)$$

$$O((R_{40} + \alpha N\frac{R_{40}}{K + 1} + (1 - \alpha)N\frac{MR_{40}}{K + 1}))I_s)$$

Putting the values $K = 3$, we get, $O(((\alpha + (1 - \alpha)M)0.25N + 1)R_{40}I_s)$. Therefore, overall computational complexity of FedORION becomes.

$$O((1 + (1 - \alpha)M)R_{40}NI_c + ((\alpha + (1 - \alpha)M)0.25N + 1)R_{40}I_s)$$

## B.2 FEDDF

FedDF performs distillation $P$ number of times at the server because it learns multiple global models.

### B.2.1 CLIENT

Considering the similar settings as FedORION, the total computation for all clients for $I_c$ iterations becomes,

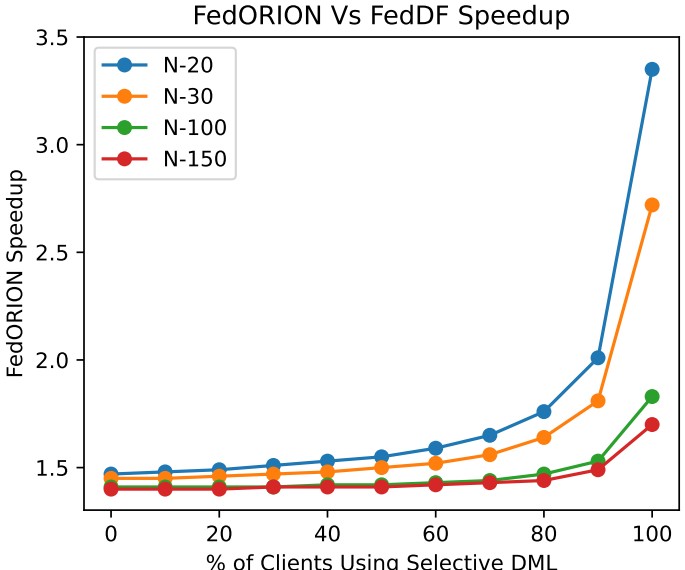

Figure 5: Speedup curve of FedORION over FedDF with varying number of clients (N) and varying percentage of clients performing SelectiveDML. For fixed number of clients, the speedup of FedO-RION increases as the clients performing selective DML increases. FedORION achieve a speedup of upto 3.25 as compared to FedDF for 20 clients, which shows the computational feasiability of FedORION over its most competitive baseline FedDF.

$$O((\alpha N R_{40} + (1-\alpha)N R_{18})I_c) = O((\alpha + (1-\alpha)M)R_{40}N I_c)$$

### B.2.2 SERVER

FedDF performs distillation for all $P$ prototypes using all of the clients models. Therefore, it performs both FP and BP for global model and only FP for rest other models. The overall computation at server for $I_s$ iterations and $P$ prototypes is

$$O(((\alpha + (1-\alpha)M)0.25N + 1)R_{40}I_s + (P-1)(((\alpha + (1-\alpha)M)0.25N + M)R_{40}I_s)$$

$$O(((\alpha + (1-\alpha)M)0.25NP + MP - (M-1))R_{40}I_s)$$

Putting the values $P = 5$ we get

$$O(((\alpha + (1-\alpha)M)1.25N + 5M - (M-1))R_{40}I_s)$$

Therefore, overall computational complexity of FedDF becomes

$$O((\alpha + (1-\alpha)M)R_{40}N I_c + ((\alpha + (1-\alpha)M)1.25N + 5M - (M-1))R_{40}I_s)$$

### B.3 COMPARING FEDORION AND FEDDF

We calculate the overall speedup as follows

$$O\left(\frac{(\alpha + (1-\alpha)M)R_{40}N I_c + ((\alpha + (1-\alpha)M)1.25N + 5M - (M-1))R_{40}I_s}{(1 + (1-\alpha)M)R_{40}N I_c + ((\alpha + (1-\alpha)M)0.25N + 1)R_{40}I_s}\right)$$

For $M$, we consider the speedup of WRN40 over ResNet18 in the ratio of their trainable model parameters i.e. $M = 11/0.5 = 22$. Additionally, putting in values, $N = 30$, and $I_s/I_c = 0.5$, We plot the overall speedup of FedORION over FedDF by varying % of clients performing selective DML and for different values of $N$ and is represented in Figure 5. We observe a considerable speedup in FedORION as compared to FedDF in heterogeneous settings. FedDF burdens the server by a considerably expensive distillation. On the other hand FedORION is efficient at the server and introduces a nominal computation at the clients. Further increasing the ratio of clients performing selective DML, FedORION can achieve even higher speedup. Moreover, FedORION outperforms FedDF by a considerable margin in terms of test accuracy over multiple datasets.

## C  ABLATION PLOTS

We show the learning behaviour of FedORION for the ablation studies performed in the section 5. Figure 6 (a) shows the importance of individual components of FedORION. Figure 6 (b) shows the behaviour of FedORION as the noise samples used for proxyless distillation are varied.

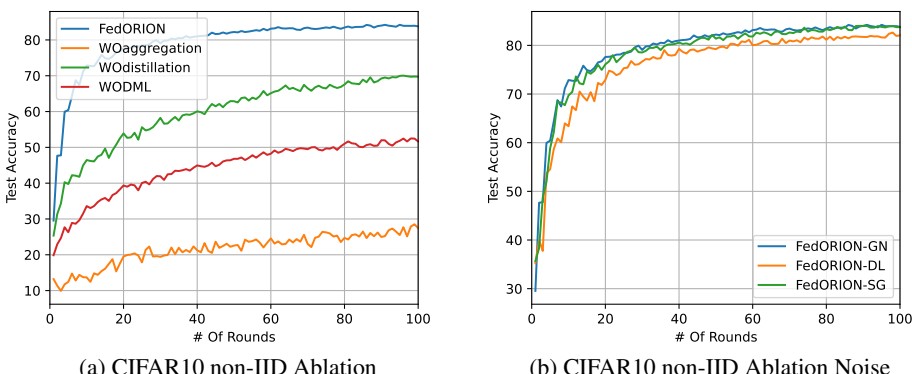

(a) CIFAR10 non-IID Ablation  (b) CIFAR10 non-IID Ablation Noise

Figure 6: Learning behaviour of FedORION in ablation studies involving ablation with various components (left) in the method and ablation with different noise samples (right) for knowledge distillation at the server side. On the left, WOaggregation, WOdistillation, WODML denotes the FedORION algorithm without aggregation, distillation, and DML phase respectively. On the right, FedORION-GN, FedORION-SG, FedORION-DL indicates FedORION algorithm with selective DML and using samples from Gaussian Noise (GN), samples generated from StyleGAN (SG) and samples from DeadLeaves (DL) for proxyless distillation respectively.

## D    COMPARISION WITH HOMOGENEOUS BASELINES

For the completeness of the experiments, we show the results by comparing with Homogeneous Baselines i.e. FedAvg, FedProx, FedNTD, and SCAFFOLD. The test accuracy on CIFAR10 dataset is reported in Table 3. FedORION shows a comparable performance with the homogeneous baselines and outperforms heterogeneous baselines on homogeneous model settings. FedNTD shows the best performance among all the methods. However, FedNTD along with the other homogeneous FL approaches like FedAvg are not applicable to model heterogeneous settings. In such settings, FedORION shows remarkable performance as compared to other baselines.

Table 3: Comparison results of FedORION and the baseline methods such as FedAvg, FedProx, FedNTD, and SCAFFOLD. The test accuracy is reported on the test set of CIFAR10 dataset.

| Methods | Test Accuracy |
|---|---|
| FedAvg | 86.3 |
| FedProx | 86.2 |
| SCAFFOLD | 85.9 |
| FedNTD | **87.9** |
| FedMD | 64.5 |
| KT-pFL | 41.8 |
| FedDF | 85.0 |
| DS-FL | 65.2 |
| FDmeta | 61.9 |
| FedKEMF | 67.3 |
| FedORION | 85.5 |

