# OpenReview forum: "FedORION: Aggregation-Assisted Proxyless Distillation for Heterogeneous Federated Learning"
_ICLR.cc/2024/Conference — ICLR 2024 Conference Withdrawn Submission_

### Official Review · Reviewer_k9TX · 2023-10-14

**Soundness:** 2 fair
**Presentation:** 2 fair
**Contribution:** 2 fair
**Rating:** 3
**Confidence:** 3

**Summary:**

This paper proposes a novel framework, FedORION, that addresses the challenge of system heterogeneity among client models using aggregation-assisted distillation without relying on the proxy/synthetic data.
However, a series of prior works have already studied KD-based proxyless distillation for heterogeneous federated learning, which are not well studied in your paper. It is important to highlight your unique contributions and distinguish them from previous methods to guarantee its solid contribution. Specifically,

1.	The mismatching problem among the client model architectures as well as the mismatch in the data has been addressed in FedDKC [1] with proxy-data-free federated distillation.
2.	The knowledge integration problem in distillation-based heterogeneous federated learning is studied in [2,3].
3.	The problem of allowing high level of system heterogeneity without public datasets is studied in [4].

[1] "Exploring the distributed knowledge congruence in proxy-data-free federated distillation." arXiv preprint arXiv:2204.07028 (2022).

[2] "A Hierarchical Knowledge Transfer Framework for Heterogeneous Federated Learning." IEEE INFOCOM 2023-IEEE Conference on Computer Communications. IEEE, 2023.

[3] "Fedgems: Federated learning of larger server models via selective knowledge fusion." arXiv preprint arXiv:2110.11027 (2021).

[4] "FedCache: A Knowledge Cache-driven Federated Learning Architecture for Personalized Edge Intelligence." arXiv preprint arXiv:2308.07816 (2023).

**Strengths:**

This paper addresses the challenge of system heterogeneity among client models using aggregation-assisted distillation without relying on the proxy/synthetic data, which is an important problem in federated learning community.

**Weaknesses:**

Actually, a series of prior works have already studied KD-based proxyless distillation for heterogeneous federated learning, which are not well studied in your paper. It is important to highlight your unique contributions and distinguish them from previous methods to guarantee its solid contribution.

**Questions:**

A series of prior works have already studied KD-based proxyless distillation for heterogeneous federated learning, which are not well studied in your paper. It is important to highlight your unique contributions and distinguish them from previous methods to guarantee its solid contribution. Specifically,

1.	The mismatching problem among the client model architectures as well as the mismatch in the data has been addressed in FedDKC [1] with proxy-data-free federated distillation.
2.	The knowledge integration problem in distillation-based heterogeneous federated learning is studied in [2,3].
3.	The problem of allowing high level of system heterogeneity without public datasets is studied in [4].

[1] "Exploring the distributed knowledge congruence in proxy-data-free federated distillation." arXiv preprint arXiv:2204.07028 (2022).

[2] "A Hierarchical Knowledge Transfer Framework for Heterogeneous Federated Learning." IEEE INFOCOM 2023-IEEE Conference on Computer Communications. IEEE, 2023.

[3] "Fedgems: Federated learning of larger server models via selective knowledge fusion." arXiv preprint arXiv:2110.11027 (2021).

[4] "FedCache: A Knowledge Cache-driven Federated Learning Architecture for Personalized Edge Intelligence." arXiv preprint arXiv:2308.07816 (2023).

---

### Official Review · Reviewer_J3qo · 2023-10-31

**Soundness:** 3 good
**Presentation:** 3 good
**Contribution:** 2 fair
**Rating:** 6
**Confidence:** 4

**Summary:**

The paper proposes a new  aggregation scheme for Federated Learning to resolve the system heterogeneity issue. The design includes the following components:
1. Noise engineered data-free distillation: On the server side, the server can perform data distillation without the assistance of public proxy data by generating noise samples as distillation samples
2. Deep mutual Learning: On the client side, the clients will have a larger local model that has distinct architecture and will be collected by server for distillation purpose and one smaller model that shares same architecture across all the clients and is collected for aggregation. The two models will mutually learn with each other.
3. Selective deep mutual learning: Considering the different clients has different resources constraints, not all clients will adopt the DML, and clients with smaller resources will only train on the model that will be sent to server for aggregation.

**Strengths:**

1. The proposed methodology addresses the issue of knowledge distillation relying on proxy data by leveraging noise samples for distillation.
2. Based on the design, on the server side, the global model can benefit from both aggregation and knowledge distillation, which can potentially improve the performance of the global model while resolving the model heterogeneous issue.
3. The ablation study thoroughly investigates how each component in the methodology will contribute to the final performance.

**Weaknesses:**

1. There are also other works contributing to the data-free distillation in Federated Learning, which are not mentioned in the paper. For example, [1] uses hyper-knowledge for data-free knowledge distillation. Although the methodology in [1] is different from FedORION, since both paper addresses the issue of relying on proxy data in KD, it is necessary for the authors to also discuss [1] or any other relevant data-free distillation in Federated Learning papers, and add them into the experiment section potentially.
2. The experiment setting that tends to mimic the system heterogeneity setting is unrealistic in the real world. The authors simply randomly assign clients with high resources (able to adopt both larger model and smaller model, WRN40 in this setting, for DML) and low resources (only able to train the smaller model, WRN40). However, in the real setting, system heterogeneity might not only distributed uniformly but also follow a "heavy-tail" distribution. More thorough discussion on mimic real-world system resource distribution can be referred in [2]. Since the paper mainly addresses the system heterogeneity issue, the reviewer would expect the experiments designed to be more aligned with real settings.
3. FedORION will be better supported if some theoretical proof exists (e.g. convergence analysis).

---
Mentioned References

[1] Chen, Huancheng, and Haris Vikalo. "The Best of Both Worlds: Accurate Global and Personalized Models through Federated Learning with Data-Free Hyper-Knowledge Distillation." arXiv preprint arXiv:2301.08968 (2023).

[2] Chen, Daoyuan, et al. "FS-Real: Towards Real-World Cross-Device Federated Learning." KDD 2023.

**Questions:**

In Table 2, I noticed that the baselines can perform much more poorly than FedORION on CIFAR100 and TinyImageNet. For FedKEMF, the final test result is only 1%, which is equivalent as random guessing. Would it be possible that these baselines can actually perform better?

**Details Of Ethics Concerns:**

None.

---

### Official Review · Reviewer_mcAL · 2023-10-31

**Soundness:** 2 fair
**Presentation:** 2 fair
**Contribution:** 2 fair
**Rating:** 3
**Confidence:** 5

**Summary:**

A solution for the model heterogeneity problem between clients is proposed in federated learning. The architecture is designed by introducing DML loss and knowledge distillation of noisy data. The experimental results show that the method is effective.

**Strengths:**

1. A federated learning framework is proposed for solving model heterogeneity.

2. The method eliminates the dependency on the publicly available proxy data by using Gaussian noise samples for performing data-free KD without compromising on the performance.

**Weaknesses:**

1. Whether the noisy data has an impact on the convergence of the local model should be analyzed theoretically.

2. It is better to add the comparison with "FedAPEN: Personalized Cross-silo Federated Learning with Adaptability to Statistical Heterogeneity" in KDD2023. The KDD method also introduces DML, which can also address the problem of model heterogeneity.

3. In the case of noniid distribution of client data, the difference of distribution between clients increases. Thus, the knowledge obtained from other clients through knowledge distillation may be a negative impact. The effect of distillation loss should be verified in more detail.

4. Some clients are randomly sampled in the experimental results of Table 1. The percentage of clients with mini-model may affect the results, and it is recommended to make relevant clarifications.

5. Overall, this paper is not innovative enough, DML is a technology that has already appeared in FedAPEN, and the distillation loss is also adopted in existing methods.

**Questions:**

1.Does the noisy data have an impact on the convergence of the local model?

2.What is the difference with "FedAPEN: Personalized Cross-silo Federated Learning with Adaptability to Statistical Heterogeneity" in KDD2023?

3.In case of client data distribution Non-IID, does the knowledge of other clients obtained through distillation can negatively impact the perfomance?

4.Why does the result of the FEDKEMF method fluctuate so much in Figure 2?

---

### Official Review · Reviewer_MRfG · 2023-11-03

**Soundness:** 3 good
**Presentation:** 3 good
**Contribution:** 2 fair
**Rating:** 5
**Confidence:** 4

**Summary:**

The paper introduces FedORION, a proxy data-free federated learning (FL) aggregation method aimed at addressing model heterogeneity. FedORION operates through Deep Mutual Learning (DML) on the client side, updating both the local and global models. On the server side, the received homogeneous global models are initially aggregated on average. The aggregated global model is further updated through knowledge distillation on Gaussian noise samples derived from the heterogeneous local models. Selective DML is also introduced, enabling DML application to specific client subsets, while clients with limited computational resources engage in traditional local training. The experimental results showcase the superior performance and robustness of FedORION.

**Strengths:**

1. Clear Description of FedORION: The paper provides a clear and understandable explanation of FedORION.

2. Interesting Experimental Results on Selective DML: The results on selective DML present interesting insights.

3. Strong Motivation: Proxy data-free federated learning plays an important role in FL.

**Weaknesses:**

1. Lack of Novelty: The paper lacks novelty as certain techniques, such as DML training on the client side and noise-based knowledge distillation, have been explored in previous works.

2. Incorrect Descriptions in Related Work: Some descriptions of related work are inaccurate and need rectification.

3. Inconclusive Experimental Results: Certain parts of the experimental outcomes, especially on the Tiny-Imagenet dataset, are not convincing due to significantly low accuracies.

**Questions:**

1.  There are lots of works [1-5] that can solve model heterogeneity (models from the same family, such as ResNet20 and ResNet56). The relevant description is incorrect in the related work section.

2. The pre-trained generator is not only used for homogeneous clients’ models, such as [4]. The relevant description is incorrect in the introduction section

3. The experimental results (Table 2) obtained on the Tiny-Imagenet dataset are significantly low and far below the reported results in the baseline papers. An accuracy of approximately 1.0% suggests that the model may not have been adequately trained.


[1] Shen, T., Zhang, J., Jia, X., Zhang, F., Huang, G., Zhou, P., ... & Wu, C. (2020). Federated mutual learning. arXiv preprint arXiv:2006.16765.

[2] Yu, S., Qian, W., & Jannesari, A. (2022). Resource-aware federated learning using knowledge extraction and multi-model fusion. arXiv preprint arXiv:2208.07978.

[3] Xia, J., Liu, T., Ling, Z., Wang, T., Fu, X., & Chen, M. (2022). PervasiveFL: Pervasive federated learning for heterogeneous IoT systems. IEEE Transactions on Computer-Aided Design of Integrated Circuits and Systems, 41(11), 4100-4111.

[4] Zhang, J., Chen, C., Li, B., Lyu, L., Wu, S., Ding, S., ... & Wu, C. (2022). Dense: Data-free one-shot federated learning. Advances in Neural Information Processing Systems, 35, 21414-21428.

[5] Yao, D., Pan, W., Dai, Y., Wan, Y., Ding, X., Jin, H., ... & Sun, L. (2021). Local-global knowledge distillation in heterogeneous federated learning with non-iid data. arXiv preprint arXiv:2107.00051.